# The Safety and Tolerability of a Potential Alginate-Based Iron Chelator; Results of A Healthy Participant Study

**DOI:** 10.3390/nu11030674

**Published:** 2019-03-21

**Authors:** Richard D. Horniblow, Pritesh Mistry, Mohammed N. Quraishi, Andrew D. Beggs, Tom Van de Wiele, Tariq H. Iqbal, Chris Tselepis

**Affiliations:** 1Institute of Cancer and Genomic Sciences, University of Birmingham, Edgbaston, Birmingham B15 2TT, UK; r.horniblow@bham.ac.uk (R.D.H.); P.Mistry.1@bham.ac.uk (P.M.); M.N.Quraishi@bham.ac.uk (M.N.Q.); A.Beggs@bham.ac.uk (A.D.B.); T.H.IQBAL@bham.ac.uk (T.H.I.); 2Queen Elizabeth Hospital Birmingham, Edgbaston, Birmingham B15 2TH, UK; 3The University of Birmingham Microbiome Treatment Centre, University of Birmingham, Birmingham B15 2TT, UK; 4CMET, Center for Microbial Ecology and Technology, Faculty of Bioscience Engineering, Ghent University, Coupure Links 653, B-9000 Gent, Belgium; Tom.VandeWiele@UGent.be

**Keywords:** alginate, iron, microbiome, ROS, prebiotic

## Abstract

Evidence supporting the ferro-toxic nature of iron in the progression of inflammatory bowel disease (IBD) is becoming well established. A microbial dysbiosis is observed in IBD patients, and intra-luminal colonic-iron is able to support a more pathogenic community of bacteria; whether this is attributed to the development of IBD and how iron could be mediating these microbial changes is still unknown. Dietary fibres are commonly used in pre-biotic supplements to beneficially affect the host by improving the viability of bacterial communities within the colon. Alginates are a class of biopolymers considered as prebiotics due to their fibre-like composition and are able to bind metal cations, in particular, iron. Considering that iron excess is able to negatively alter the microbiome, the use of alginate as a food supplement could be useful in colonic-iron chelation. As such, this first-in-man study aimed to assess whether the use of alginate as a dietary iron chelator was both safe and well tolerated. In addition, the impact of alginate on the microbiome and iron levels was assessed by using an intestinal model SHIME (Simulation of the Human Intestinal Microbial Ecosystem). Alginate was supplemented into the diets (3 g/day) of healthy volunteers (*n* = 17) for 28 days. Results from this study suggest that daily ingestion of 3 g alginate was well tolerated with very minor side effects. There were no detrimental changes in a variety of haematological parameters or the intestinal microbiome. The bacterial communities within the SHIME model were also not influenced by iron and or alginate; it is possible that alginate may be susceptible to bacterial or enzymatic degradation within the gastro-intestinal tract.

## 1. Introduction

It has recently been highlighted that elevated concentrations of iron within the colon are associated with the progression and development of intestinal diseases, and specifically inflammatory bowel disease (IBD) [1,2,3]. This is supported by the finding that depleting iron within the colon is associated with an ameliorated phenotype [4]. Additionally, enzymes responsible for the breakdown of iron-complexes (heme-oxygenases) were found to be upregulated and protective in instances of inflammation within the colon, possibly in an attempt to diminish iron’s pro-inflammatory nature [5]. High levels of colonic luminal iron have been shown to (i) be pro-inflammatory [6,7], (ii) have catalytic reactive oxygen species (ROS) generating ability [6,8], (iii) negatively alter the microbiome [1,9,10,11,12,13] and (iv) have cellular proliferation effects [14,15]. When dietary iron consumption is in excess of nutritional demands (average consumption of 10 to 12 mg with only 2 mg required) and only 0.7 to 22.9% of ingested iron is absorbed [16], this suggests a high residual concentration of iron in the colon. This is particularly the case for patients with IBD who often present with iron-deficiency anaemia and are on oral iron supplements [17,18].

An intervention that would render this excess colonic iron inert would theoretically be highly beneficial; a class of bio-actives that may achieve this would be non-absorbable iron chelators. To date, there have been a number of reports that detail the iron-chelating effects of potentially non-digestible fibres, of which sodium alginate is of particular interest [19,20,21]. Alginates are natural polysaccharides extracted from the cell walls of brown seaweeds. Chemically, these biopolymers are formed from unbranched (1–4) linked β-D-mannuronic acid (M) and its C5 epimer, α-L-guluronic acid (G) [22]. Alginates are used within the food industry as gelling agents and in anti-reflux preparations due to their ability to gel, forming “rafts” within the stomach to suppress acid-reflux [23]. Most recently, an alginate of relatively short chain length and high M content (MLD) has been identified as an ideal iron chelator, demonstrating limited affinity towards calcium [24]. MLD is a 145 kDa biopolymer with a G:M ratio of 38:62 which is able to bind ferrous iron (0.6 mg Fe/mg MLD) and inhibit murine intestinal iron absorption [24]. Previous clinical studies have demonstrated the ability of alginate supplementation to decrease iron absorption and hence reduce serum-iron concentrations [25]. The use of MLD may be of particular use in the treatment of gastro-intestinal diseases associated with high levels of unabsorbed, luminal iron within the colon. No human studies have been conducted to assess the tolerability of MLD consumption in healthy individuals over a period of 28 days.

Thus, the overall aim of this study was to assess the safety, tolerability and feasibility of 3 g daily alginate consumption in healthy volunteers over a period of 28 days. This was predominantly examined using intestinal health questionnaires, in addition to haematological and faecal microbiome assessments. In a similar approach, the impact of MLD on iron levels and the microbiome was assessed in a highly controlled in vitro model of the human colon. Such results will guide future clinical assessments utilising MLD in iron-chelation therapies.

## 2. Materials and Methods 

### 2.1. Healthy Participant Study Design

Seventeen healthy individuals, over the age of 18 years of either sex, with no previous history of gastro-intestinal disease were recruited between October 2015 and November 2015. Informed consent was obtained for all participants recruited. Participants completed a quality of life questionnaire (Short Inflammatory Bowel Disease questionnaire (S-IBDQ) (McMaster University, Ontario, Canada) (Appendix A)) and provided a stool sample before consuming MLD. Answers were scored 1–7, with 1 indicating large changes in lifestyle/general wellbeing and 7 indicating no changes or problems. Participants were provided with a diary to record compliance, monitor changes in bowel habit/frequency and document the presence of any adverse symptoms for the duration of the study. Appointments were made for the subsequent two visits required at the midpoint (day 14) and end (day 28) of the study. At each of these visits, further blood and stool samples were collected and S-IBDQ completed. The study design is summarised in Appendix A. Venous blood was taken for analysis of full blood count and biochemistry (Appendix A).

### 2.2. Ethics Statement

The study was approved by the National Research and Ethics Service Committee West Midlands on 20 August 2015 (REC reference: 15/WM/0221). The study was registered at www.isrctn.com as ISRCTN16202716. 

### 2.3. Faecal Microbiota Assessment (Healthy Participant Study and SHIME)

For the assessment of changes in the colonic microbiome, DNA was extracted from stool using the QIAamp DNA stool mini kit (QIAGEN, Hilden, Germany) following the manufacturer’s protocol. Three replicate polymerase chain reactions were performed for each faecal sample. In brief, the V4 region of the 16S rRNA gene was amplified with region-specific primers that include the Illumina flowcell adapter sequences. After cluster formation on a HiSeq/MiSeq instrument, the amplicons were sequenced with custom primers. 

Quality filtering of reads was applied. Sequences were clustered into operational taxonomic units (OTUs) with 97% similarity and picked using open reference OTU reference picking protocol with a 97% similarity threshold. Taxonomic assignments of OTUs that reached the 97% similarity level were made using QIIME (Quantitative Insights Into Microbial Ecology) by comparison with the Greengene (http://rdp.cme.msu.edu/) databases (gg_13_5_otus). Alpha-diversities of the gut microbial communities were compared by calculating the number of observed OTUs, Shannon Diversity and Faith’s Index, and Chao index. Beta diversity was calculated by both unweighted and weighted Unifrac distances calculated. Differences in OTU abundances were tested with ANCOM (Analysis of Composition of Microbiomes). To predict functions encoded by the genomes of bacteria, we performed a phylogenetic investigation of communities by reconstruction of unobserved states (PICRUSt) analysis based on the 16S rRNA analyses. Estimated abundances of Kyoto Encyclopedia of Genes and Genomes (KEGG) Orthology groups were compared between different time points. We obtained 21520246 high-quality filtered reads, corresponding to an average of 377,548 reads per subject. Contaminant reads were negatively filtered, and triplicates were collapsed by the mean value for each observation mean. Reads were clustered into 3525 operational taxonomic units (OTUs) at 97% sequence identity following which their representative sequences were used in taxonomic analysis. All sequences were classified from phylum to species.

### 2.4. Healthy Volunteer Faecal Iron and Calcium Assessment

Two pools of iron and calcium were assessed; (i) total iron/calcium and (ii) “free” iron/calcium. The total iron/calcium in each sample represents the total metal content, regardless of coordination to ligands/proteins/chemicals whereas the “free” pool represents iron and calcium that is non-associated. To determine the total iron/calcium pool, faecal samples (3 g, wet weight) were ashed in silica crucibles at 480 °C for 48 h within an open-air furnace. The ashed product was re-suspended in 0.1 M HCl (5 mL) and stored. To probe the “free” pool of metal, faecal samples (3 g, wet weight) were homogenised in DI H_2_O (3 mL) until a slurry was obtained. An additional volume of DI H_2_O (7 mL) was added to the slurry and centrifuged at 7500× *g* for 30 min. The supernatant was collected, and the protein content was removed by acidification. Iron and calcium concentration within the samples was determined using flame atomic absorption spectroscopy (fAAS). Water content of faecal samples was determined by drying overnight at 70 °C and further desiccation.

### 2.5. Faecal Reactive Oxygen Species Assessment

Faecal samples (2 g, wet weight or 3 mL SHIME colon material) were homogenised and subsequently incubated for 18 h in TRIS buffered saline (4 mL, pH 7.0) containing DMSO (5% (*v*/*v*)), glucose (0.1% (*w*/*v*)) and EDTA (50 mM) under agitation at 37 °C. Faecal slurries were then centrifuged and the supernatant acidified using HCl to precipitate the protein, which was further centrifuged and the resulting supernatant pH re-adjusted to pH = 7.0. Samples were stored at this point at −20 °C until batch analysis could be performed with all samples. Standards were made to run alongside the samples. These were methansulfinic acid (0–10 mM) dissolved in the TRIS buffered used to homogenise the faecal samples; samples and standards were processed identically. A 1.3 mL aliquot of the sample was mixed with H_2_SO_4_ (10 M, 250 μL) and centrifuged at 3500 RPM for 5 min. The supernatant (1.4 mL) was mixed thoroughly with 1 M sulphuric acid saturated 1-butanol (4 mL) and centrifuged at 3500 RPM for 5 min to allow phase separation. The upper phase was aspirated (3.4 mL) and mixed with sodium acetate (0.5 M, pH 5.0, 2 mL) and again centrifuged at 3500 RPM for 5 min to allow phase separation. The lower aqueous phase was removed (1.8 mL) and pH adjusted to pH 2.5, before the addition of FastBlue BB salt (14 mM, 800 μL) and incubated for 15 min in the dark. After this period, a toluene:butanol (3:1) mixture (1.5 mL) was added and thoroughly mixed for 2 min, before phase separation by centrifugation at 3500 RPM for 5 min. The upper phase (1 mL) was removed and washed with 1-butanol saturated water (2.0 mL). Finally, the samples were centrifuged at 3500 RPM for 5 min, the upper layer (1 mL) was aspirated, mixed with a pyridine:acetic acid (95:5) mixture (100 μL) and absorbance read at λ = 415 nm.

### 2.6. Intestinal Model (SHIME)

A M-SHIME model was set up as previously described, with proximal colonic vessel attachments only [26]. In brief, the model consisted of a stomach/small intestinal (SI) vessel (not inoculated with faecal bacteria) which fed three independent colonic vessels (inoculated with faecal bacteria) which were regulated at pH 5.6–6.9 (proximal colonic environment). Colonic vessels were adapted with mucin-beads for surface-attached microbes. The beads (K1-carrier, AnoxKaldnes AB, Lund, Sweden) were prepared by boiling mucin and agar in water until they formed a gel and suspended within the colonic vessels.

Each treatment within this experiment (Fe only, MLD+Fe and control) had individual stomach/SI and colonic (*n* = 3) vessels. Colonic vessels were inoculated, simultaneously with faecal bacteria obtained from one healthy donor. Before supplementing with Fe or MLD, the colonic vessels were equilibrated and established over a 7-day period; between day 7 (*t* = 0) and 22 (*t* = 15), the colonic vessels were supplemented with either iron (15 mg) or MLD (1.0 g three times daily). Colonic vessels were flushed daily with N_2_ to maintain anaerobicity, and bacterial integrity and health were monitored by measuring short chain fatty acids (SCFAs) and ammonium concentrations regularly. The protocol for these assessments has been previously described [27]. All experiments were run in parallel simultaneously. The whole system was maintained at 37 °C.

The supplementation of aqueous Fe and MLD into SHIME was performed through the addition of stock solutions into the stomach vessel; three injections of FeSO_4_ in 0.1 M HCl (0.5% *w*/*v*, 1 mL) and three injections of MLD (4% *w*/*v*, 12.5 mL) per day. The control vessel received DI H_2_O only and volumes were adjusted accordingly to ensure each vessel received equal volumes of DI H_2_O. These concentrations were pre-optimised to obtain a total Fe concentration within the colonic vessels of c. 400 µM and a free Fe concentration of 100 µM. These concentrations were chosen based on previous in vitro studies examining the iron chelation potential of MLD, and only FeSO_4_ was used as the iron source in these experiments.

### 2.7. SHIME Iron Assessments

The measurement of total Fe in the colonic vessels was undertaken by aspirating samples (200 μL) of colonic vessels and mixing with 200 μL of TCA (20% *w*/*v*) solution. These were boiled for 5 min at 100 °C and subsequently spun at 1500 RPM for 5 min to pellet debris. The supernatant (200 μL) was aspirated and mixed with a ferrozine solution (600 μL). Measurement of free Fe levels was performed by aspirating 1 mL of colonic medium and spun at 1000 RPM for 1 min to pellet the bacteria. The supernatant (500 μL) was aspirated and directly added to the top of a 30,000 MWCO spin column. This was spun at 15,000 RPM for 10 to 15 min, until equal amounts of filtrate were collected and used directly with the ferrozine reagent (600 μL), as previously described.

### 2.8. Thin Layer Chromatography (TLC)

TLC was performed to assess the integrity of the alginate following culturing within the colonic vessels of SHIME. A positive control for alginate degradation was prepared by heat degradation as reported previously [24], and degradation derivatives were assessed using a orcinol-based staining reagent, as detailed previously [28].

### 2.9. Statistical Methods

All analysis was performed using Microsoft Excel. Single factor ANOVA was used to detect changes in the blood results. Paired two-tailed Student’s *t* test was used to detect changes in the levels of bacteria when comparing each of the three time-points in the study at a significance level of 0.05. Values are stated as mean ± standard deviation or median (range).

## 3. Results

### 3.1. Study Participant Baseline Demographics

All recruited individuals (13 female and 4 male participants, with a median age of all participants of 42 ± 9.5 years) completed the study. MLD was well tolerated with no serious side effects reported. Minor adverse effects were documented (Table 1), with the predominant complaint of excess flatus followed by bloating.

There were small but significant changes in S-IBDQ participant survey scores when comparing baseline to day 14 and study endpoint. Mean scores at baseline were 65.5 ± 2.9, which decreased to 62.2 ± 4.0 (*p* = 0.013) at day 14 and 62.2 ± 4.6 (*p* = 0.018) at day 28; the total maximum score was 70 (Figure 1A).

Upon consideration of the individual participant responses, these significant changes were associated with questions 4 and 6; 4. How often during the past two weeks have you been troubled by pain in the abdomen? 6. Overall, in the past 2 weeks, how much of a problem have you had with passing large amounts of gas?

The mean response value for question 4 was 6.8, 5.4 and 5.6 at baseline, day 14 and day 28, respectively (*p* < 0.005). These scores equate to a drop from “None of the time” to “A little of the time” or “Hardly any of the time”. For question 6 the mean response was 6.6, 4.3 and 4.4 at baseline, day 14 and day 28, respectively (*p* < 0.001). equating to a drop from “No trouble” to “Some trouble”. The questionnaire and associated scores are detailed in Appendix A.

### 3.2. Haematological Biochemical Analyses

Haematological and biochemical analyses were assessed over the period of the intervention. There were no significant changes in measured levels of haemoglobin (Figure 2A), creatinine (Figure 2B), Calcium (Figure 2C), alanine aminotransferase (Figure 2D), and magnesium (Figure 2E).

### 3.3. Faecal Iron and Calcium Concentration

Total iron and calcium concentrations from day 0, 14 and 28 remained constant (Figure 3A/C). Average total and free iron concentrations across all samples were 0.37 ± 0.17 and 0.014 ± 0.02 mg/g faecal dry weight, respectively. There were no significant changes in mean free iron values from day 0 to day 14 and 28 (Figure 3B). Average total and free calcium concentrations across all samples were 323.4 ± 380.5 and 36.66 ± 15.15 mg/g faecal dry weight, respectively (Figure 3B/D). No statistical changes in the calcium pools measured were detected (Figure 3D).

### 3.4. Faecal Reactive Oxygen Species

Over the time-course of the intervention there were no significant changes in faecal ROS generating potential, with mean MeSH concentrations of 1.8 ± 0.6, 2.23 ± 1.23 and 1.98 ± 0.64 mM/g dry weight faecal material detected at day 0, 14 and 28, respectively (Figure 4).

Correlations between ROS generating potential and concentrations of total/free iron and calcium were also assessed (Appendix A); no significant associations were found. No correlation was found between calcium concentrations and ROS generating potential in any of the pools assessed (Appendix A).

### 3.5. Healthy Participant Microbiome Analysis

The four major gut phylae showed very similar taxonomic compositions. Although the proportion of Bacterioides increased (Δ21) and Firmicutes (Δ21) reduced by day 28 compared to baseline, this was not significant. No significant changes at the family, genus and species levels were observed between any time points.

Both species richness and diversity were similar for the different sampling time points. There was no significant change in the gut microbial community structures and individual OTUs when compared between any time points.

### 3.6. Iron, ROS and Ammonia Concentrations within SHIME

To assess in a more controlled environment the impact of MLD on intestinal iron levels an artificial gut model (SHIME) was employed. Total iron changes within the colonic vessels were assessed over the period of the iron/alginate supplementation (day 0, 6 and 15). All iron-fed vessels were established to have an equal and constant iron concentration throughout the period of the study. However, small but significant iron concentration differences were measured for Fe and MLD+Fe vessels at baseline (353 and 400 μM, respectively) and midpoint (395 and 356 μM, respectively) (Figure 5A).

Free iron changes were additionally measured (Figure 5B). However, no significant differences were measured between the Fe and MLD+Fe throughout the study period. With total iron concentrations changing within the vessels, the free iron measured was normalised with the total iron, and a percentage of free iron with respect to total iron was calculated (Figure 5C); no significant percentage differences were calculated over the study.

ROS concentrations were also measured throughout the study with no significant changes between the treatment (Fe and MLD+Fe) and control colonic vessels. Additionally, ammonium concentrations for day 0, 6 and 15 of the study are also reported (Figure 5D).

### 3.7. SHIME SCFA Analysis

Short chain fatty acid (SCFA) assessments were carried out routinely as a measure of bacterial viability throughout the study (Appendix A). Relative SCFA concentrations were calculated with respect to total SCFA changes. Overall concentrations of butyric acid concentrations were elevated with MLD+Fe treatment with respect to Fe and control, with statistical significance found at day 6 and day 13 (Appendix A).

### 3.8. SHIME Microbiome Analysis

No differences in alpha or beta diversity in the microbial profiles between the three treatment arms and time points were observed. Additionally, there were no significant differences identified in OTUs between the groups.

### 3.9. Breakdown of Alginate

Alginate breakdown derivatives were assessed using TLC (Figure 6). As a positive control, MLD was degraded by heat, from which degradation products were observed (Figure 6A); increased streaking and higher retention factors (RFs) were observed for longer degradation times. The possibility of degradation of alginate within the colonic vessels of SHIME was also assessed (Figure 6B). SHIME medium from the colonic vessels supplemented with MLD+Fe (at the end of the experiment) demonstrated extensive streaking indicative of degradation, though, of note, the control samples with no alginate similarly demonstrated a streaking profile albeit less pronounced.

## 4. Discussion

Iron excess within the colon has been demonstrated to negatively regulate the microbiome, influencing the development of pathogenic communities [9,11,13,29,30,31]. The mechanism behind this toxicity is not fully understood, but it is likely due to one of two factors: (1) An abundance of iron promotes the growth of iron-requiring pathogenic bacteria [9] or (2) chemically, it is able to modulate the communities through toxic reactions [6,30,32]. An intervention which would evade both possible mechanisms of microbial dysbiosis would be to chelate the excess iron, rendering it inert, non-reactive and non-utilisable by bacteria. Dietary iron chelators, of which one particular biopolymeric alginate (MLD) was assessed in this first-in-man study, with the primary aim of ascertaining the safety and tolerability associated with consumption. This was assessed through the use of questionnaires and various biomarkers of health including a variety of haematological parameters and changes to the microbiome. In addition, assessments were also undertaken using SHIME to assess the impact of alginate supplementation in a highly controlled intestinal model.

The consumption of alginate in this study was 3.0 g per day, which is greater than that consumed in an average diet (of which there are many estimations, but maximum daily exposure is estimated at 2.1 g daily) [33]. In the healthy participant study, alginate supplementation was well tolerated with little to no side effects observed. However, there were small decreases in the overall S-IBDQ score following alginate consumption. Upon interrogation of the individual questionnaire responses, participants reported an increase in abdominal pain from “none of the time” to “hardly any of the time” and from having problems passing large amounts of gas from “no trouble” to “some trouble”. The reason for these changes could be an increase in bacterial fermentation-processes, presumably induced by MLD [34]. This is supported by the observations reported in the artificial gut of a trend for increased butyrate production following alginate supplementation. These physiological changes in the amount of bloating and gas are important to consider in the context of the cohort of patients that such therapies are aimed at, namely those with IBD. As discussed, removal of excess free iron from the bowel of IBD patients would be highly advantageous; however, if the bioactive increased the frequency of the symptoms associated with this disease, the use of such an agent needs to be carefully considered. A limitation of this study was the small cohort size (*n* = 17); the completion of this initial safety assessment of MLD allows for both larger test cohorts and testing in IBD patients in future studies.

To further assess the safety of alginate supplementation, a variety of haematological parameters were assessed. Reassuringly, 28-day consumption of alginate did not reduce the haemoglobin levels, and neither were there any effects on liver biomarkers. In addition, there were no changes in calcium concentration which is important since alginates have a strong affinity for calcium [35]. Since the intestinal microbiome is integral to health and a dysbiosis has been linked with a variety of diseases, alterations in bacterial communities were also assessed following alginate supplementation. Throughout the duration of the study, minimal changes were observed with no net positive or negative growth effects on the bacterial species present. While a shift in the composition of communities present may have been expected with alginate consumption, no such changes were observed. This may be due to the fact that the participants were “free living” with no dietary restrictions. In addition, there inevitably would have been gross differences in diets consumed across all 17 of the participants. Any possible changes induced by alginate consumption may have been overwhelmed by changes influenced by these dietary differences.

These limited changes in the human study are in agreement with the lack of changes observed in the gut model with MLD. Interestingly in this artificial gut model, there were also no alterations in the microbial communities when colonic vessels were co-incubated with a concentration of c. 400 to 600 µM total iron and c. 200 to 300 µM free iron (based on a 70% iron binding capacity by SHIME medium). This is surprising since previous studies have linked iron excess with dysbiosis. The lack of a bacterial dysbiosis in the colonic model could be due to one of several reasons. First, ferrous iron was the only form and species of iron utilised in the SHIME studies, and it could be the case that other forms of iron within the diet cause greater dysbacteriosis. This highlights the importance of understanding the form of iron likely to be present throughout gastrointestinal transit. Second, the SHIME feed contains many iron chelating additives (such as pectin and mucin) [19,36,37] which may both alter the availability of iron to specific bacterial communities and reduce the catalysis of ROS production (and indeed, no increased ROS production was found in iron-supplemented colonic vessels compared to control). The list of chemical contents within the SHIME feed is listed in Appendix A. Third, the concentration of iron (15 mg/day) may not have been sufficient to induce significant changes. An additional reason as to why limited microbiome changes were found could be due to the iron-species present within the SHIME gut model in comparison to that found in man. Specifically, the results of this study may highlight the importance of dietary ligands on iron and how they impact on the form of iron present upon entering the colon; iron-speciation can radically alter its physico-chemical properties (for example, comparison of the reactivity of a “free” ferrous iron to ferric-oxides) [6,8] and, as such, may alter its activity within a model intestine [38]. The limited changes in ROS concentrations between the iron and control treatments provide evidence to support this, where oxidised and hydrolysed iron forms are unable to partake in redox-reactions. Finally, and most interestingly, is the argument that in the absence of the host compartment (i.e. the colon), iron is unable to mediate any microbiome changes. This may indicate that the observed dysbiosis reported in the literature may be a consequence of alterations in the host which subsequently trigger shifts in the microbiome (in an indirect mechanism). If so, the use of SHIME would only detect a direct action of iron in modulating the microbiome, which was not observed here. This may be a key factor in understanding the mechanism by which iron can induce a microbial dysbiosis within the colon. Such non-direct effects on the microbiome have been reported for other chemicals such as chemotherapeutics [39].

With no changes incident upon iron supplementation within SHIME, a reversal of dysbiotic shifts during co-administration of a fibre-like iron chelator (MLD) would not be observed; this was indeed the case. It is considered that MLD could act as a prebiotic and as such, bacterial changes would be expected, however, no changes were found in this study. This is surprising since the flatulence and bloating which was reported by the healthy volunteers following alginate consumption likely reflects fermentation of the alginate and would, thus, likely result in a change in the microbiota to efficiently utilise this dietary source [40,41]. The degradation experiments and the trend for increased butyrate production, indeed, likely points to fermentation of the alginate. This is further supported by a previous study which reports that alginate is degraded within the colon [40].

Thus, in summary, MLD ingestion was safe and well tolerated, with relatively minor side effects reported of increased flatulence and bloating. Furthermore, there was no evidence of any detrimental changes to the microbiome. This was somewhat surprising, since the reported increased flatulence and bloating likely reflects alginate fermentation and this would likely result in some changes to the composition of the microbiome. To separately assess the impact of iron on the microbiome and whether this could be reversed using an iron chelator, an artificial gut model was utilised, and the results of this were equally surprising with no apparent change in the microbiota with iron and or alginate. This likely reflects the other iron chelating agents found within the food matrix and the need to characterise the exact species of iron in the colon that is responsible for the dysbiotic effects previously reported. It is evident from these studies that any future applications of using alginates as luminal iron chelators require a formulation where bioactivity is targeted to the colon and protected from any dietary agents and/or digestive processes to maximise iron-binding activity.

## Figures and Tables

**Figure 1 nutrients-11-00674-f001:**
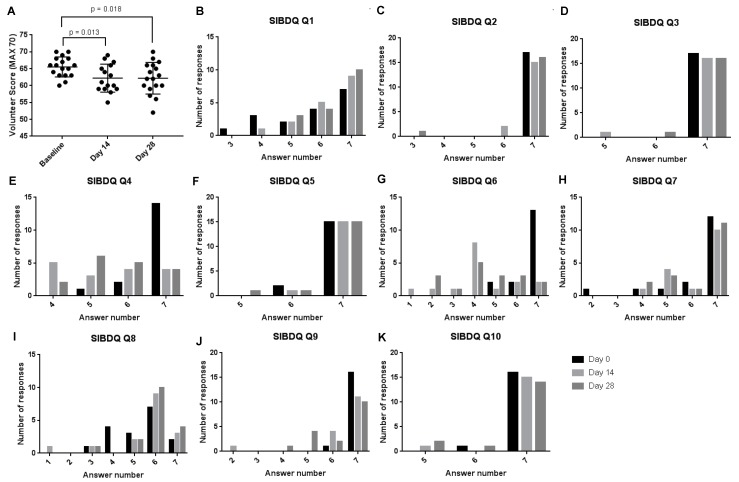
(**A**) S-IBDQ-score obtained from volunteer questions at baseline, day 14 and day 28. Each point represents a volunteer score with the midline and standard deviations plotted. Absolute histograms (**B**–**K**) of individual patient responses for each of the S-IBDQ questions (1–10 respective) at day 0, 14 and 28. Given *p* values represent statistical significance using an un-paired *t*-test with *p* < 0.05 denoting statistical significance.

**Figure 2 nutrients-11-00674-f002:**
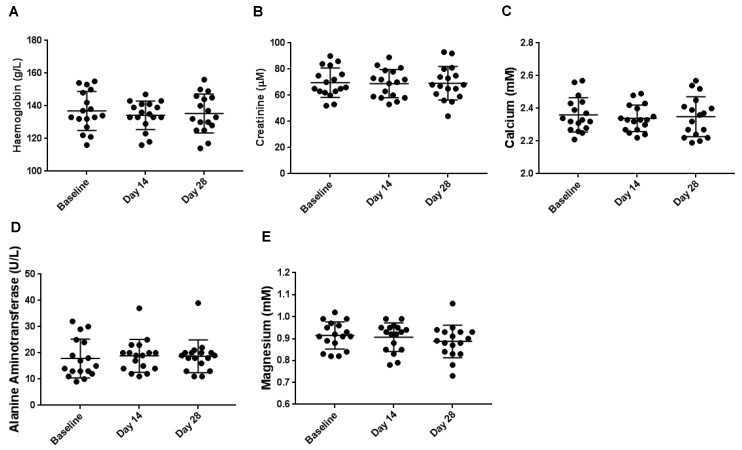
Haematological measurements for (**A**) haemoglobin, (**B**) creatinine, (**C**) calcium, (**D**) alanine aminotransferase and (**E**) magnesium. Each point represents a healthy participant with the midline and standard deviations plotted. 1way ANOVA tests of significance were performed across all time points, and no significance was observed.

**Figure 3 nutrients-11-00674-f003:**
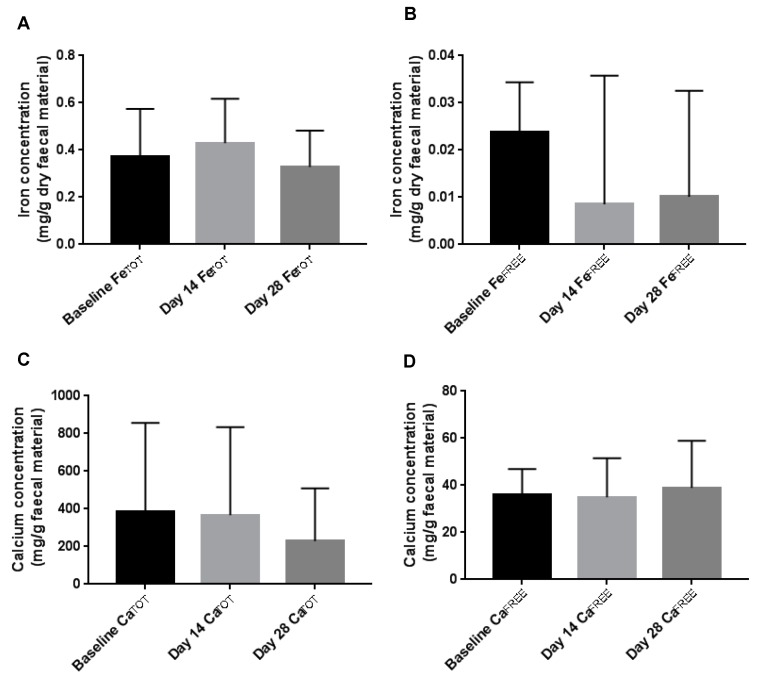
Faecal iron and calcium concentrations (per gram dry weight faecal material) at baseline, day 14 and day 28 of the intervention. (**A**) Total iron, (**B**) free iron, (**C**) total calcium and (**D**) free calcium. Mean values are plotted with error bars denoting standard deviations in the error. No statistical significance (using 1way ANOVA tests) was found between the data sets.

**Figure 4 nutrients-11-00674-f004:**
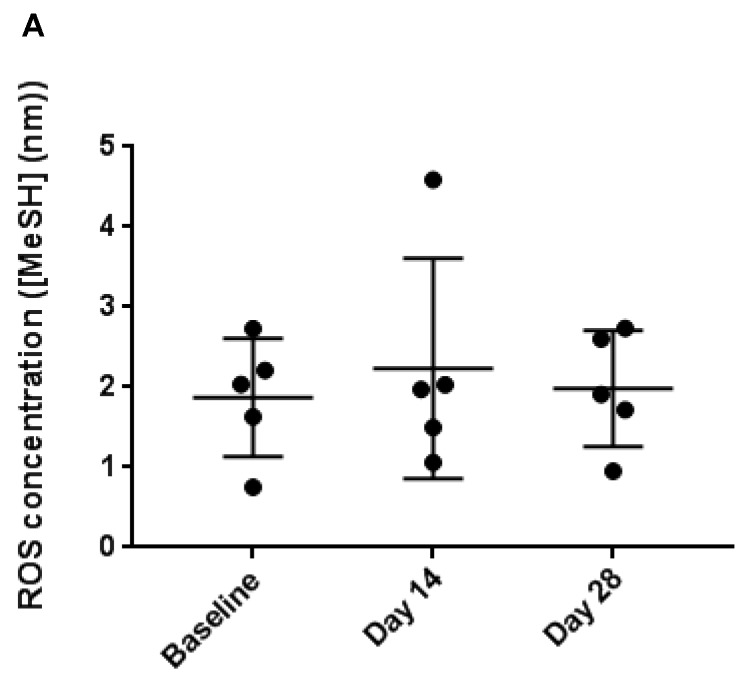
Faecal reactive oxygen species (ROS) concentrations at baseline, day 14 and day 28 of the intervention. No significant changes were observed (1way ANOVA with *p* < 0.05) across all time points.

**Figure 5 nutrients-11-00674-f005:**
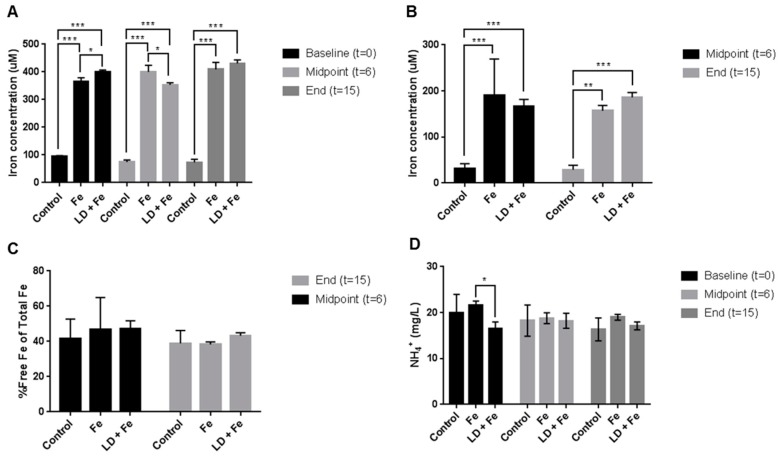
Iron and ammonium parameters within SHIME colonic vessels. Total iron (**A**) was measured across three time points (baseline, midpoint and at the end of the study) in the control, iron (Fe) and iron with MLD (MLD + Fe) vessels (*n* = 3). Free iron (**B**) was also measured at the midpoint and the end. The percentage of free iron with respect to total Fe was also calculated (**C**). Ammonium concentrations were also measured across three time points (**D**). Bars represent mean values with error bars denoting standard deviations in the error. Statistical significance (using 1way ANOVA tests) is represented with an *, ** or **, where *p* < 0.05, 0.005 and 0.0005 respectively, with *n* = 3 for each mean value.

**Figure 6 nutrients-11-00674-f006:**
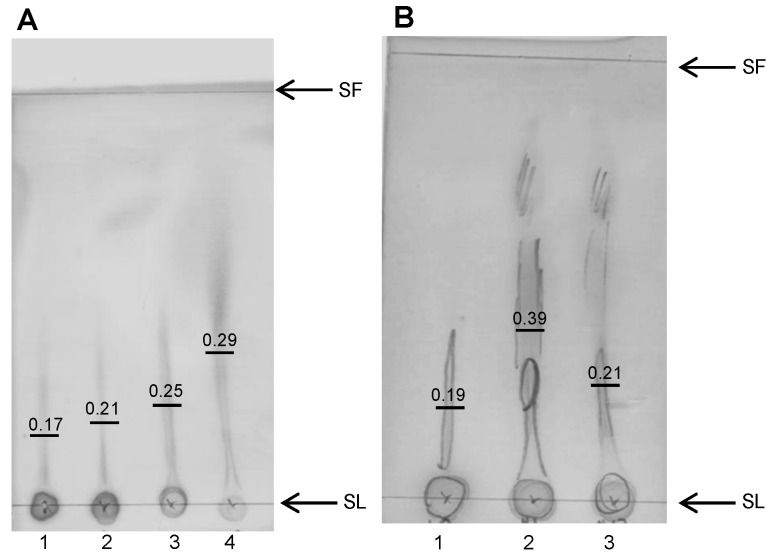
Thin layer chromatography (TLC) plates spotted with (**A**) 1: Native MLD, 2: 5 min heat degraded MLD, 3: 15 min heat degraded MLD, 4: 60 min heat degraded MLD and (**B**) 1: Native MLD, 2: Day 16 MLD colonic vessel and 3: Day 16 control colonic vessel. Solvent Fronts (SF) and Spotting Lines (SL) are indicated. Retention values are calculated and represented on the TLC with a midline.

**Table 1 nutrients-11-00674-t001:** Table of reported adverse effects over the 28-day intervention.

Adverse Effect	Number of Participants Reported
Flatulence	16
Bloating	7
Heartburn	2
Abdominal Pain	8
Loose Motion	4
Urgency to Defecate	1
Nausea and Vomiting	3
Headache	1

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
