# Peer review of "The Safety and Tolerability of a Potential Alginate-Based Iron Chelator; Results of A Healthy Participant Study"

_nutrients, 2019, doi:10.3390/nu11030674_

Reviewer 1 Report

Based on previous observations that in several intestinal models the MLD preparation of sodium alginate inhibited cellular iron accumulation and suppressed iron absorption in mice, the authors studied the safety and tolerability of the preparation on healthy human subjects. The results show that the preparation was well tolerated and safe, giving just minor inconveniences, and that it did not affect fecal microbiome and iron status. They also used the intestinal model SHIME to verify the effect of iron and MLD, but surprisingly they found that neither had evident effect on the intestinal microbioma. The work is carefully done, well organized and well presented, although the results are mostly negative. 

- The lack of any evident effect of the alginate on total and free iron in the faecis and in the SHIME model argues for a weak, if any, iron binding activity of the MLD.  Thus, probably naming it “iron chelator” is not really appropriate,  particularly in the title.

- More details on the MLD molecule structure should be given, which include the molecular weight, the binding capacity of ferrous and ferric iron.

- The finding that iron addition did not trigger shifts in the microbiome in the M-SHIME experiments led to the conclusion that the iron-induced dysbiosis reported in literature is not due to iron itself by to indirect mechanisms. This is only speculative, since only one type of iron was used, the formulation of which was not given.   The iron treatments in the SHIME experiments should be described more in detail.

Author Response

RESPONSE TO REVIEWER 1

Many thanks for your insightful comments on our manuscript.   We have amended the manuscript to reflect your suggestions and believe it is now markedly improved.  Please see below a point-by-point breakdown of your comments and how the manuscript has been amended.

Comment #1:             “The lack of any evident effect of the alginate on total and free iron in the faeces and in the SHIME model argues for a weak, if any, iron binding activity of the MLD.  Thus, probably naming it “iron chelator” is not really appropriate, particularly in the title.”

Amendment: We agree that the evidence presented in this study suggested limited iron-chelating capacity of Manucol LD. However, based on previous literature and our own historic studies demonstrating the affinity of Manucol LD towards iron, we have replaced ‘novel’ with ‘potential’ to reflect this.  We have now changed the title such that it reads:

‘THE SAFETY AND TOLERABILITY OF A POTENTIAL ALGINATE-BASED IRON CHELATOR; RESULTS OF A HEALTHY PARTICIPANT STUDY’.

We also feel it is of particular importance with regards to the remit of the special edition that we are submitting to. However, if the reviewer feels strongly about this we can remove any reference to iron chelation.

Comment #2:             “More details on the MLD molecule structure should be given, which include the molecular weight, the binding capacity of ferrous and ferric iron.”

Amendment: We agree that these are important details to include. As such, the manuscript has been updated to include the following:

“MLD is a 145 kDa biopolymer with a G:M ratio of 38:62 which is able to bind ferrous iron (0.6 mg Fe/mg MLD) and inhibit murine intestinal iron absorption.[24]”

We are unfortunately unable to report a ferric-iron binding capacity due to its reactive nature and limitations when handling experimentally.  Furthermore, we are unaware of the exact form and species of iron within the colon. Thus we have solely quoted the ferrous iron binding capacity, which relates to our previous in vitro studies.

Comment #3:             “The finding that iron addition did not trigger shifts in the microbiome in the M-SHIME experiments led to the conclusion that the iron-induced dysbiosis reported in literature is not due to iron itself by to indirect mechanisms. This is only speculative, since only one type of iron was used, the formulation of which was not given.   The iron treatments in the SHIME experiments should be described more in detail.”

Amendment: This is an important comment, and as such, the SHIME experiments involving iron have been edited to include more details. It now reads as such:

In the Materials and Methods:

“The supplementation of aqueous Fe and MLD into SHIME was performed through the addition of stock solutions into the stomach vessel; three injections of FeSO4 in 0.1 M HCl (0.5 % w/v, 1 mL) and three injections of MLD (4 % w/v, 12.5 mL) per day. The control vessel received DI H2O only and volumes were adjusted accordingly to ensure each vessel received equal volumes of DI H2O. These concentrations were pre-optimised to obtain a total Fe concentration within the colonic vessels of c. 400 µM and a free Fe concentration of 100 µM.  These concentrations were chosen based on previous in vitro studies examining the iron chelation potential of MLD, and only FeSO4 was used as the iron source in these experiments”

In the Discussion:

“The lack of a bacterial dysbiosis in the colonic model could be due to one of several reasons. Firstly, ferrous iron was the only form and species of iron utilised in the SHIME studies, and it could be the case that other forms of iron within the diet cause greater dysbacteriosis. This highlights the importance of understanding the form of iron likely to be present throughout gastrointestinal transit.”

Reviewer 2 Report

Dear Sir

the study entitled THE SAFETY AND TOLERABILITY OF A NOVEL ALGINATE-BASED IRON 3 CHELATOR; RESULTS OF A HEALTHY PARTICIPANT STUDY by Horniblow et al investigates the effects of alginate in healthy volunteers and in a Simulator of the Human Intestinal Microbial Ecosystem. Both alginate and Iron do not influence the studied biomarkers and alginate is in general well tolerated

The study is very interesting and well written. Methods are adequate to the aim and the results clearly shown

I have no criticisms

Author Response

RESPONSE TO REVIEWER 2

Many thanks for reviewing our manuscript and your positive comments. No edits have been made in response to your review.

Reviewer 3 Report

Aim of the present study was to assess safety and  tolerability of alginate consumption in healthy volunteers.

The study is welll conducted. However, it suffers from some limitations:

- the number of volunteers enrolled ( n= 17) is extremely small to draw statistically significant conclusions. If the Authors are not in the possibility to enlarge the cohort, the must clearly state this limitation in the Discussion section

- why not to use a blinded control group with palcebo administration?

- The Authors state in the Results section that side effect were infrequent; however, data reported show the presence of flautulence in 16/17 subjects (that is 94% !) and of bloating and abdominal pain in about 40-50% of subjects. Could the Authors explaine why these side effects were defined as "infrequent"?

- Adverse effects describied could be grater in IBD patients which, as reported by the Authors in the Introduction, represent the category of patients who could theoretically benefits form treatment with iron chelators. Please make a comment regarding these aspects.

Author Response

RESPONSE TO REVIEWER 3

Many thanks for your insightful comments on our manuscript detailing where there are limitations.   We have amended the manuscript to reflect your suggestions and believe it is now improved.  Please see below a point-by-point breakdown of your comments and how the manuscript has been amended.

Comment #1:             “The number of volunteers enrolled (n= 17) is extremely small to draw statistically significant conclusions. If the Authors are not in the possibility to enlarge the cohort, the must clearly state this limitation in the Discussion section”

Amendment: We agree that the size of the cohort was small in this feasibility study and unfortunately we are unable to enlarge the group size. We have made it explicit in the discussion that this is a limitation, and suggested that any future studies should be undertaken with more participants.  The edit now reads:

“A limitation of this study was the small cohort size (n=17); the completion of this initial safety assessment of MLD allows for both larger test cohorts and testing in IBD patients in future studies.”

Comment #2:             “why not to use a blinded control group with placebo administration?”

Response: It would have been ideal to undertake such clinical study; however, due to the small recruitment number it was unfeasible. This will certainly be taken into consideration in future studies utilising MLD (now safety has been verified).

Comment #3:             “The Authors state in the Results section that side effect were infrequent; however, data reported show the presence of flatulence in 16/17 subjects (that is 94% !) and of bloating and abdominal pain in about 40-50% of subjects. Could the Authors explain why these side effects were defined as "infrequent"?

Amendment: We regret this oversight in our manuscript. This may be confusing as the table reports number of patients who experienced such effects over the whole intervention period. To avoid any future confusion, we have removed the word ‘infrequent’.

Comment #4:             “Adverse effects described could be grater in IBD patients which, as reported by the Authors in the Introduction, represent the category of patients who could theoretically benefits form treatment with iron chelators. Please make a comment regarding these aspects”

Amendment: We agree with this statement and believe it is important to raise the point within the discussion. As such, the edit now includes:

“These physiological changes in the amount of bloating and gas are important to consider in the context of the cohort of patients that such therapies are aimed at, namely those with IBD.  As discussed, removal of excess free iron from the bowel of IBD patients would be highly advantageous; however, if the bioactive increased the frequency of the symptoms associated with this disease, the use of such an agent needs to be car
